# Alarm Calling in Plateau Pika (*Ochotona curzoniae*): Evidence from Field Observations and Simulated Predator and Playback Experiments

**DOI:** 10.3390/ani13071271

**Published:** 2023-04-06

**Authors:** Meina Ma, Rui Hua, Darhan Bao, Guohui Ye, Zhuangsheng Tang, Limin Hua

**Affiliations:** College of Grassland Science, Gansu Agricultural University, Key Laboratory of Grassland Ecosystem of the Ministry of Education, Engineering and Technology Research Center for Alpine Rodent Pests Control, National Forestry and Grassland Administration, Lanzhou 730070, China

**Keywords:** plateau pika, alarm call, acoustic analysis, playback experiment, vigilance response

## Abstract

**Simple Summary:**

With the continuous progress in research technology, it is found that auditory signals are particularly important for social rodents. The plateau pika *(Ochotona curzoniae)* is a type of rodent with song characteristics. The study of the biological meaning of its song can provide a theoretical basis for understanding the adaptive evolution of the communication methods of burrowing rodents. So far, the acoustic signal and behavioral response analysis of the plateau pika have been carried out independently, and there is no comprehensive understanding of the biological significance of the calls of the plateau pika. In this study, based on previous studies on the call of plateau pikas, it is assumed that plateau pikas can use acoustic signals to transmit alert information. To test this, we first collected the alarm call of plateau pikas, and then used field playback experiments to verify the alarm call of plateau pikas. Finally, in order to determine the universality of the alarm call of plateau pikas, we conducted playback experiments in different regions. Our research results show that plateau pikas can transmit alert information through acoustic signals, and their alert calls are responded to differently in different regions.

**Abstract:**

Acoustic communication plays a vital role in passing or sharing information between individuals. Identifying the biological meaning of vocal signals is crucial in understanding the survival strategies of animals. However, there are many challenges in identifying the true meaning of such signals. The plateau pika (*Ochotona curzoniae*) is a call-producing mammal endemic to the Qinghai–Tibet plateau (QTP) and considered a keystone species owing to its multiple benefits in alpine rangeland ecosystems. Previous studies have shown that plateau pikas emit alarm calls as part of their daily activities. However, only field observations have been used to identify these alarm calls of the plateau pika, with no attempts at using playback experiments. Here, we report the alarm calling of plateau pikas through field observations as well as simulated predator and playback experiments in the Eastern QTP from 2021 to 2022. We found that both female and male adults emitted alarm calls, the signals of which comprised only one syllable, with a duration of 0.1–0.3 s. There were no differences in the characteristics between the observed alarm calls and those made in response to the simulated predator. The duration of the alarm call response varied with altitude, with plateau pikas living at higher altitudes responding at shorter durations than those at lower altitudes.

## 1. Introduction

Intra- and interspecific communications are important aspects in the daily activities of animals in the wild [1,2] since they perceive different stimuli from the surrounding environment. One of their survival strategies is largely dependent on how effectively they can transmit and share information among heterospecific or conspecific individuals [3,4]. Indeed, each animal species has a unique communication strategy to meet its specific needs corresponding to the prevalent environmental circumstances and restrictions. This communication strategy involves sharing sensory cues through various communication signals including visual, auditory, olfactory and tactile communications [2,5]. Among the different types of interspecific signals, acoustic communication is a more efficient communication signal than other communication signals [6]. Compared with visual, olfactory and tactile signals, the characteristics of sound signals can also be competently changed in terms of duration, phonation frequency and syllables at the same time. They can also carry a large amount of information, are not limited by type of terrain, and can be transmitted faster and over a long distance [7].

Worldwide, rodents are the most diverse of the small mammals [8], and are mostly cave-dwelling. Previous studies have shown that rodents rely mainly on olfactory communication [9], with acoustic communication having little or even no biological significance [10]. However, with continuous advancement in research techniques, it is now clear that audible signals are particularly important for social and gregarious rodents [11]. They play an important role in territorial behavior, courtship, warning and other activities by emitting different sounds [12]. For example, Pearson’s tuco-tuco *(Ctenomys pearsoni)* will frequently make a unique courtship sound in the initial stage of reproduction [13]; when gerbils *(Rhombomys opimus)* are frightened, they will emit a short whistle to alarm other individuals [14]; and squirrels of different species can emit different alarm calls, with some even possessing a variety of different alarm sounds to transfer information about different predators [15]. Alarm calls are an important aspect of animal communication that provide early warning to individuals regarding external threats [16]. The individual that first perceives the danger among social animals sends a warning signal to other individuals, so that the group can enter an alert state and choose to stop activities and/or escape quickly from the scene. Animal vigilance is considered to be an important means to obtain social recognition and detect predators [17], which is of paramount importance for the safety of the whole group.

Grassland rodents usually choose to live in groups to reduce the risk of predation and the consumption of energy through thermoregulation [18]. This social lifestyle favors the usage of vocal communication more predominantly over other ways that transmit information over longer ranges and consequently reduces energy expenditure. On the Qinghai–Tibet plateau (QTP), the plateau pika *(Ochotona curzoniae)* is a rodent-like small mammal inhabiting the alpine rangeland ecosystem at an altitude of 3000–5100 m [19]. Pikas can make long sounds to declare sovereignty after territorial invasion [20]. The structure of the pika’s cochlea is better adapted than other rodents within the same habitat for receiving high-frequency acoustic signals [21].

The study of acoustic communication in the plateau pika started in 1986. To date, there have been four papers that have reported the types of calls and their meanings in plateau pikas. Smith et al. [22] firstly defined seven types of calls in plateau pikas through field observations; the seven types of calls are long call, short call, tone shifting call, mournful call, trembling call, muffled call, and sharp call, revealing that the male adult emits long calls and females short calls. The short calls most likely encompass alarm calls in response to approaching raptors, yaks or humans. However, this work relied on the researchers’ observations in the field; there were no playback experiments to test the short calls that were recorded. He et al. [23] analyzed the acoustic characteristics of calls in plateau pikas and speculated on their biological meaning, ranging from courtship, to raising alarm, to scaring away predators. In another study, Hua et al. [12] concentrated on the temporal distribution of long calls and their influencing factors in male plateau pikas. In addition, when Zhang et al. [24] studied the predation risk of plateau pikas, it was found that they were highly sensitive to predator “voices” and the distressed screams of other plateau pikas. However, the characteristics of risk-related signals emitted by plateau pikas were not analyzed, and the correspondence between these signals and the behaviors displayed was ignored.

From the above studies, it is clear that the analysis of acoustic signals and behavioral responses in plateau pikas have thus far been independently conducted, and a comprehensive view of the biological meanings of pika calls is lacking. Therefore, the aim of this study was to identify the alarm calls of plateau pikas via an integrated approach that recorded short calls and their accompanying behavioral responses in the field and used a simulated predator to induce the alarm calls. Furthermore, playback experiments were conducted to verify the reliability of the biological meanings of alarm calls in plateau pikas. 

## 2. Materials and Methods

### 2.1. Study Sites

Three sites located in Bola township (34°90′92″ N, 102°76′91″ E; altitude: 3212 m), Maai township (34°54′44″ N, 102°15′38″ E; altitude: 3383 m) and Nima township (34°00′41″ N, 102°03′65″ E; altitude: 3527 m), in Xiahe county, Luqu county and Maqu county, respectively, were employed in this study (Figure 1). All three of these counties are located in the Eastern QTP. The grassland type at the three sites is alpine meadow, with the dominant species being *Kobresia humilis*, *Elymus nutans*, *Potentilla fragarioides*, *Potentilla anserina* and *Poa annua* (Table 1). The three sites have a typical alpine continental climate with a long cold season and short warm season.

At the Bola township study site, we recorded the pika calls and their accompanying behaviors as the pikas emitted the calls. We also used a simulated predator to induce the plateau pika into emitting alarm calls and conducted playback experiments. At the other two study sites, we also conducted the playback experiments to verify the reliability of the alarm calls obtained in the field. 

### 2.2. Definitions of Vigilance Behavior and Call Types in Plateau Pikas

The complexity of the calls along with the researchers’ previous delimitation of the seven types of calls were too subjective to serve as a basis for classification. The plateau pika’s call is temporarily only divided into long and short calls based on the duration of the call. The calls of plateau pikas were classified into two types (long and short) according to the call duration. Long calls were more than 10 s and short calls less than 10 s. We adopted the definitions of vigilance behavior and call types in plateau pikas outlined by Smith [25] and Zhang [24]. Specifically, the plateau pika’s vigilance behavior involves suddenly sitting on the grass with its neck straight, suddenly ceasing to eat, standing and watching, and quickly fleeing to their hole or cave when the pika receives the warning signal. 

### 2.3. Call Recordings and Behavioral Observations in the Field

Plateau pikas are diurnal small mammals with a relatively greater seasonal activity in March and April than in early spring. Previous research has shown that only adult plateau pikas utter long and short calls in spring [20]. Using the focal and scan sampling method [26], we recorded the calls of plateau pikas and their accompanying behaviors in March and April 2021 in the Bola township. For clarifying the different calls emitted by female and male individuals, we captured 20 pikas as target animals, including 5 females and 15 males. Animal marker solutions in red and bule were used to mark the female and male individuals, respectively. After marking the animals, we returned them to their habitat. A Sony A7R3 video camera (SONY^®^, Tokyo, Japan) connected to an SY-322 shotgun microphone (RODE^®^, Australia, Sydney) was used to record both the calls and the behavior of the marked individuals in their habitat. Recordings were made for four hours in the morning and four hours in the afternoon, for 10 days in each month, resulting in a total of 280 h of recordings in March and April 2021. The purpose of the recordings was to select the potential alarm calls from all the short calls emitted by plateau pikas.

### 2.4. Alarm Call Recordings and Identification with the Simulated Predator

For identifying the real alarm calls of plateau pikas, we built a weasel-like, remote-control piece of equipment that we used to simulate a predator to induce the marked pikas into producing alarm calls. All observers stood far out of sight of the marked pikas. One of the observers controlled the equipment to move forward toward the marked pikas, while the other observer recorded the alarm calls induced by the simulated predator and the pikas’ accompanying behavior. 

Adobe Premiere Pro 2020 software was used to select the alarm calls with obvious vigilance behavior and turn them into .wav format. Then, we used Adobe Audit 2020 software to reduce the noise and analyze the available acoustic signals to obtain the sound duration, sound frequency and number and interval of syllables [20]. 

### 2.5. Playback Experiment

A Bluetooth speaker (BV660, China, Shenzhen) placed in the center of the pikas’ habitat was used to play the alarm calls processed by the software in the field. The loudness of the alarm call playback was 88 dB. A Sony A7R3 video camera was positioned alongside the speaker on the ground. The observers remained inactive in the field for 15 min prior to the start of the playback experiment and, after the pikas had adapted to the presence of the observers [27,28], we remotely controlled the speaker to play the processed alarm call (the duration of one minute) and the camera then recorded the vigilance behavior of the pikas around the speaker. In May 2022, we made the alert call into a long audio frequency with a duration of 15 min, and added a 2 s blank tone between the two adjacent alert calls as an interval. A three-day field playback experiment was conducted in Xiahe county, Luqu county, and Maqu county at different altitudes to observe and record whether plateau pikas still provide vigilance behavior feedback and vigilance behavior duration. 

### 2.6. Data Analysis

The video recordings were imported from the video camera in .mp4 format into a notebook computer. First, we used Adobe Premiere Pro 2020 software to select sound clips with obvious vigilance behavior and turn them into the required .wav audio format. Then, we used Adobe Audit 2020 software to reduce the noise and intercept the audio files that needed to be analyzed to obtain the sound duration, sound frequency and the duration and intervals of syllables in the sound. Open Adobe Audit 2020 software was used to import the target audio and obtain the waveform diagram and spectrum diagram of the target audio. Frequency analysis was selected in the configuration menu window option, in which the higher the FFT size, the larger the analysis resolution. Higher FFT sizes report frequency data more accurately, but they require longer processing time. Hamming and Blackman options that can accurately reflect the center frequency were considered, and the number of syllables in each sentence were counted via the language map of Adobe Audit 2020. The measurement parameters included sentence duration, syllable duration and syllable interval. The gene frequency and frequency range of each song were counted using the frequency spectrum. All data were recorded in Excel 2019, and the mean square deviation of the duration of different vocal segments and syllable intervals were calculated in SPSS19.0 software.

## 3. Results

### 3.1. Call Types of Plateau Pikas

During the experiment at the Bola township site in April 2021, a total of 201 effective calls were recorded, including 157 long calls and 44 short calls. These long calls were emitted by male pikas; however, the short calls were emitted by both male and female pikas (Figure 2). The duration of the long calls ranged from 10 s to 26 s (Figure 3A), while that of a short call was less than 10 s. Long calls consisted of more than 20 syllables, whereas short calls comprised one or fewer than 20 simple syllables. In a short call, there was a very short chirp with a duration of 0.1–0.3 s (Figure 3B) and composed of only one syllable. There were many kinds of accompanying behavior for long calls, while short calls were mostly accompanied by warning behaviors. When one plateau pika emitted a short chirp, others nearby performed vigilance behavior (Figure 4).

### 3.2. Preliminary Judgments of Alarm Call

Through field video observation, it was found that plateau pikas produced no obvious response to the arrival of cattle and sheep on the grassland, and remained in whatever their previous state was at the time. However, if their companions emitted a very short chirp, the observed plateau pikas would immediately show behaviors such as sitting on the grass with their necks stretched out, stopping their feeding, standing and watching, quickly escaping to the burrow entrance, or entering the burrow (Figure 5). These behaviors all have warning significance. At this moment, a very short call of this type accompanying the vigilance behavior was defined as a quasi alarm call.

### 3.3. Sound Characteristics Accompanying Warning Behavior

In the field, the remote-control model was used to simulate a nearby disturbance from a predator. It was found that plateau pikas would make a very short call when faced with an approach of the remote-control model, and the other pikas would quickly enter an alert state. When the sound was emitted, the animal that made the sound would shake their body with the sound, and then the other animals would stop eating, stand and watch, quickly escape, or enter their burrow. This kind of chirp was the same as the short chirp alert issued by plateau pikas in a natural and non-interfered environment. It is a short, single-tone chirp with only one simple syllable. The sound parameters of the two chirps are shown in Figure 6.

### 3.4. Playback Experiments and Alarm Call Response

After recording and observing the vigilance behavior of wild plateau pikas, the alarm calls of individuals that showed obvious vigilance behavior were recorded and extracted after noise reduction. Four temporary alarm calls in 2021 were selected, and five in 2022, resulting in a total of nine sounds. Each sound was verified by field playback experiments, five times at random. The results showed that plateau pikas respond vigilantly to the alarm calls played in the field (Figure 7), but their responses to the same alarm call was different between the different locations. The alarm calls recorded in Xiahe county were processed into 15 min audios and played back at different altitudes in Xiahe county (altitude: 3212 m), Luqu county (altitude: 3383 m) and Maqu county (altitude: 3587 m). Although the plateau pikas in these various locations displayed vigilance behavior in response to the alarm calls, the duration of the vigilance behavior differed. Specifically, the duration of alert behavior followed the order Xiahe county (altitude: 3212 m) > Luqu county (altitude: 3383 m) > Maqu county (altitude: 3527 m) (Figure 8).

## 4. Discussion

He et al. [23] proposed that the calls of plateau pikas with different acoustic characteristics represent different biological meanings, but did not determine or verify them. Wang and Smith [28] suggested that only male plateau pikas emit long sounds, while females and males can emit short calls, which may contain alert implications, but no further analysis was made in this regard. The present study found that plateau pikas can transmit alert information by emitting a very short chirp with a duration of less than 2 s. This was uncovered through observation and analysis, and then the function of this short chirp was verified experimentally. Our result differed from the analysis by Hua [12] with regard to plateau pikas’ chirps, who concluded that alarm calls were long chirps. We found that vigilance behavior was always accompanied by a simultaneous short chirp. Therefore, based on the current research, it is believed that short chirps are more consistent with an alarm call in plateau pikas. Plateau pikas are primary consumers in the ecosystem [29] and are at the bottom of the food chain. In the face of a large number of predators, pikas need to remain alert whilst taking in energy sources [30]. Compared with other communication types, vocal communication is more conducive to transmission in an open environment, effectively improving the communication efficiency between populations [31], and it is of great significance to identify, evaluate and take behavioral decisions on potential risks [32]. Compared with other methods, plateau pikas tend to use sound signals to transmit warning information. Unlike the long calls that only males can make, short calls comprise a kind of chirping that both females and males can make. At the same time, unlike long calls, which increase significantly in the breeding period, short calls are made in both the breeding and non-breeding periods. Considering the predation status and the high energy consumption of the tweeting of the plateau pika on the QTP, in order to reduce the risk of predation as much as possible and adapt to extreme weather to maintain energy, the plateau pika should choose short and high frequency tweets when expressing alerts. If the chirping time is too long, the risk of exposure will be increased during the chirping process. At the same time, tweeting is a high energy consumption behavior. An alert based on the predatory status of the plateau pika is a frequent behavior. If the alert is a constant tweeting, it requires a greater energy supply, which is not conducive to the survival of these animals in extreme weather. Therefore, a short call is more suitable than a long call for the plateau pika to transmit an alarm signal.

Animal acoustic communication research methods combined with functional anatomy can help gauge the functional relationships between animals’ vocal organs, signal-receiving organs, acoustic parameters and individual differences. For example, it was found that the structure of the cochlea in plateau pikas is highly adapted to receiving high-frequency acoustic signals, as compared to the same structure in the subterranean plateau zokor *(Eospalax Fontanierii)* [21], although they inhabit the same area. The ethological significance of auditory communication can be inferred from the responses of other individuals, i.e., by indicating the biological importance of the sound. For instance, through the playback calls of the newborn cubs of captive giant panda *(Ailuropoda melanoleuca)*, the interactive behavior of the mother was observed, which determined the sound function of the cubs [27]. In order to better determine the warning call of plateau pikas, this study combined sound signals and a behavioral analysis; that is, the acoustic characteristics of the call were analyzed and the signal function determined through the reaction of the animal. In previous studies on acoustics and behavior, telescopes were often used for behavioral observation, and a hand-held recorder was connected to a microphone for sound recording, which made the audio and pictures independent of each other and out of sync, while the recordings of behavior could not provide a reference in the latter stage. By comparison, the present study adopted an integrated approach to recording by connecting the camera with the telephoto lens and the directional microphone in the natural state of the field. The telephoto lens ensured sharpness in the long-distance recording and reduced the level of interference of human observation on the object. This integrated connection setup of the microphone and camera improved the consistency between the speaker and the observer during the recording process. When using this instrumental configuration to produce recordings, a complete set of video data for auxiliary reference when analyzing and judging the voice function in the latter stage could be provided. As for the determination of the voice function, previous research has mostly replied on the subjective judgment of human observers. In our study, the warning function was verified through a variety of methods such as natural collection analysis, simulated natural enemy collection, and return field visits, so as to reduce the level of human interference and retain the natural state of the field as much as possible, thereby improving the accuracy of the voice function determination.

During the present study, it was observed that plateau pikas produced no obvious response to the arrival of cattle and sheep on the grassland, but would immediately respond with vigilance to an unknown presence, such as that simulated by our remote-control model. It is likely that, as part of the long-term survival process, pikas have learned that livestock pose little to no threat, and hence show no obvious vigilance behavior in their presence. The remote-control model was a strange and unknown presence to the plateau pika, meaning that the perceived risk would be high and they would immediately enter an alert state to ensure their safety once the model appeared. Note that we chose to use a ground-based remote-control model instead of an unmanned aerial vehicle (UAV) because the former is easy to camouflage, is relatively quiet, and is easier to operate and control. Conversely, with UAVs, which are noisy, it is impossible to distinguish whether the plateau pika’s vigilance behavior derives from the threat of a perceived predator or simply from the considerable noise that the UAV produces. The use of UAVs for simulating an airborne natural enemy is not yet mature, because these types of predators in the sky will not make a loud noise when hunting.

During the playback experiments, it was observed that plateau pikas would respond with vigilance to the playback of alarm calls at different times and that the alarm calls could also elicit vigilance responses in different areas. This supports our observation that alarm calls are common in plateau pika populations, rather than a special behavioral adjustment or response between specific individuals at a specific time or geographical location. During the study, plateau pikas responded differently to different alarm calls, which may be because there was too much background noise in the field during the sound processing, and the pursuit of low noise in the noise reduction process would distort some of the audio, resulting in differences in the results of the playback experiments. In the field playback experiments, windless and sunny weather was selected as much as possible to eliminate the interference of wind speed and direction on sound transmission, but the existence of natural wind is still inevitable. Therefore, the behavioral response of plateau pikas to different alarm calls may be affected by wind speed and direction, resulting in observed differences when playing back the sound [33].

It was found that plateau pikas in different regions produced vigilance behavior when facing the same alarm call, but the duration of the vigilance behavior differed. This may be because plateau pikas in different places have “dialects” when they chirp, which makes their voices different. When the recorded alarm calls were played back in the sound acquisition areas, plateau pikas responded with long-term vigilance behavior. In a study of the naked mole (*Heterocephalus glaber*), it was also found that unique group dialects could be detected, and it was shown via audio playback that individuals gave priority to their original group dialect [34]. The ability of pikas to understand and react depends on both their survival instinct and experience from past threats [35]. One possible explanation for the difference between the playback vigilance responses in different locations is the different altitudes of the three regions, as this may cause disparities in sound transmission such that the response duration of plateau pikas to the same alarm call may vary. Because sound waves are mechanical waves that need some kind of medium to spread, conditions such as higher altitude, thinner air, and lower conductivity reduce the effect and distance of sound propagation [36]. The greater the energy loss of the audio signal with the same frequency and loudness in the transmission process, the smaller the transmission distance and effect, and the shorter the vigilance time displayed by plateau pikas. Moreover, with rising altitude and thinner air, plateau pikas are likely to change their sound characteristics and emit different “dialects” in order to adapt to the local living environment and reduce energy consumption. This also shows the importance of vocal communication to the survival of wild rodents.

The ability of plateau pikas to survive relies on the timely discovery of the existence of risk, making quick and effective strategies, and regulating the time and energy allocation of different behaviors. The plateau pika has become an important part of the alpine meadow grassland ecosystem on the QTP because of its fast reproduction and strong adaptability [37], which has had multiple impacts on the grassland ecosystem. The rapid growth of plateau pika populations has broken the original ecological balance, resulting in the fragmentation of the grassland landscape and a decline in grassland productivity [38]. At present, rodent prevention and control methods using drugs are often used, which focuses on reducing the density and total number of rodents by eliminating the species from the local ecosystem within a certain period of time and in a particular region [39]. However, such an approach ignores their function and role in the grassland ecosystem, thus destroying the original constraint relationship of the grassland ecosystem, which is not conducive to the maintenance of biodiversity and creates more favorable habitat conditions for the occurrence of rodent damage. Recently, international pest control has turned to the EBSPC (ecologically based strategies for pest control) [40,41], which pays attention to basic ecological research on pests and the importance of species in an ecosystem, and comprehensively considers the population size, risk level, prevention and control costs and other factors [42]. In other words, it uses ecological principles to achieve control objectives [43]. Through the study of rodent ecology, it has been found that the best period to control grassland rodent damage is the rodent breeding period [44,45,46]. According to the basic viewpoint of optimality theory, natural selection will encourage animals to make use of time and energy allocations in the most effective manner to carry out a variety of activities, thus allowing them to reach an optimal state [43]. Potentially, we could take advantage of the influence of predation risk on reproductive behavior by emitting a simulated alarm call to plateau pikas to create a risky environment during their breeding period, thereby altering the allocation of time and energy in their reproductive behavior, which might lead to disturbance in their reproductive activities and thus influence their population characteristics. Such an approach might be useful in studying and controlling plateau pika populations on the QTP.

Due to time limitations, the present work can only be considered to be a preliminary study on alarm calls among plateau pikas. Owing to the complexity and variability of other sounds, it is difficult to determine their biological meaning in such a short period. Moreover, when the biological meaning of the emitted sound is uncertain, it is difficult for verification experiments to achieve meaningful results, and so more time is needed to collect a larger number of calls for analysis.

## 5. Conclusions

Plateau pikas can use acoustic signals to transmit information, and different calls have different acoustic characteristics and represent different biological meanings. The alarm call of plateau pikas is a very short call. In the face of an alarm call issued by different individuals at different times, vigilance behavior can be elicited in other individuals as a response. Plateau pikas in different regions still respond with vigilance behavior in response to the same alarm call, but their alarm call may have dialect differences, and individuals will give priority to the dialect unique to their original location.

## Figures and Tables

**Figure 1 animals-13-01271-f001:**
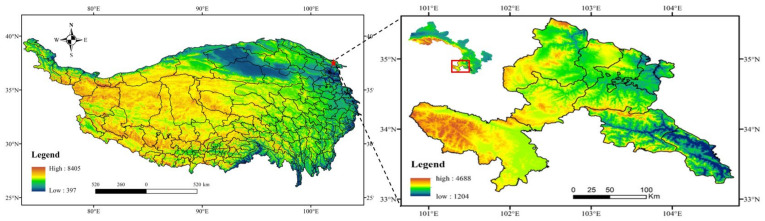
Location of the study area and sites in China.

**Figure 2 animals-13-01271-f002:**
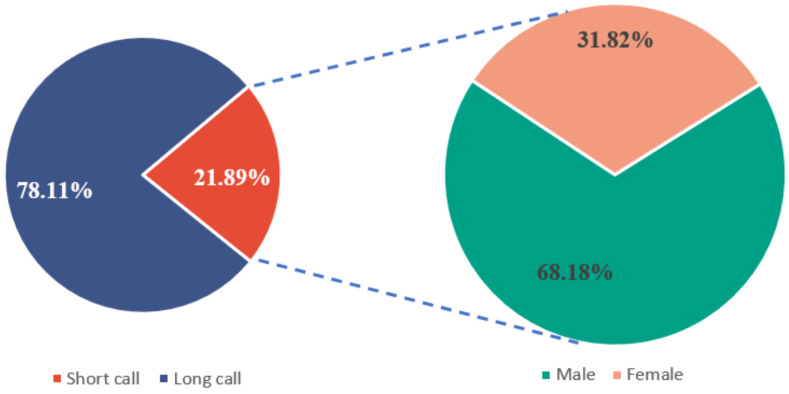
The proportion of males is higher for long calls, while the proportion of females is higher for short calls. Through further statistical testing (*p* < 0.05), it was found that there was a significant difference in long and short calls between the sexes.

**Figure 3 animals-13-01271-f003:**
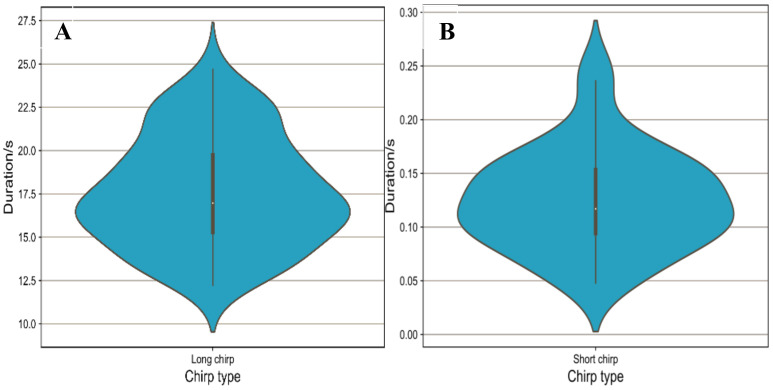
Ranges of long (**A**) and short (**B**) calls of plateau pikas. In the violin plots, the lower and upper edges of the box represent the 25% (q1) and 75% (q3) quartiles, respectively. The white dot inside the box represent the median (md).

**Figure 4 animals-13-01271-f004:**
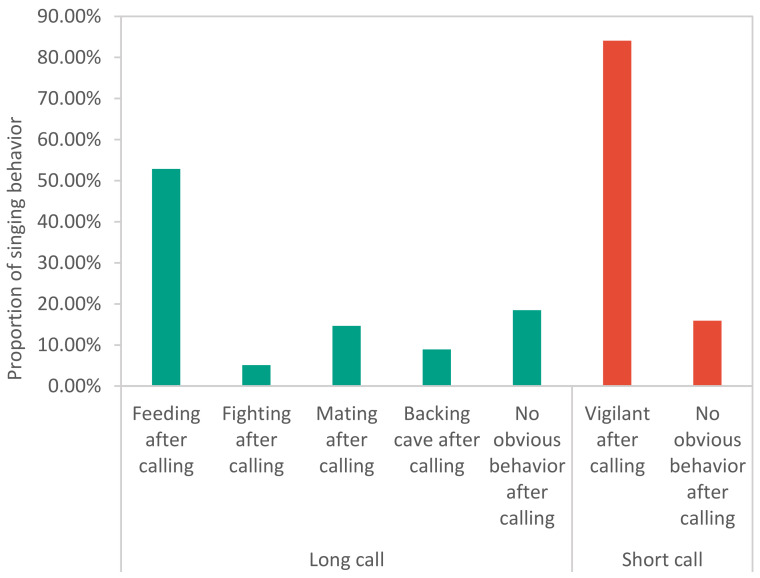
For long calls, feeding, fighting, mating, retreating to the burrow and no obvious behavior after calling accounted for a relatively high proportion, whereas vigilance after calling accounted for a high proportion for short calls. Through further statistical testing (*p* < 0.05), it was found that there were significant differences between long and short calls in terms of the different behaviors after calling.

**Figure 5 animals-13-01271-f005:**
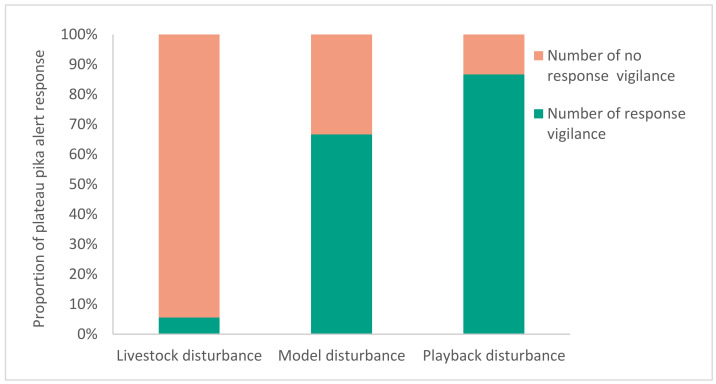
The stimulation experiment is divided into three types (livestock, model, playback). The number of livestock tests was *n* = 18 (9 consecutive days, once in the morning and once in the afternoon); the number of model tests was *n* = 15 (3 continuous days, 5 times a day at random); and the number of playback tests was *n* = 15 (3 consecutive days, 5 times a day at random).

**Figure 6 animals-13-01271-f006:**
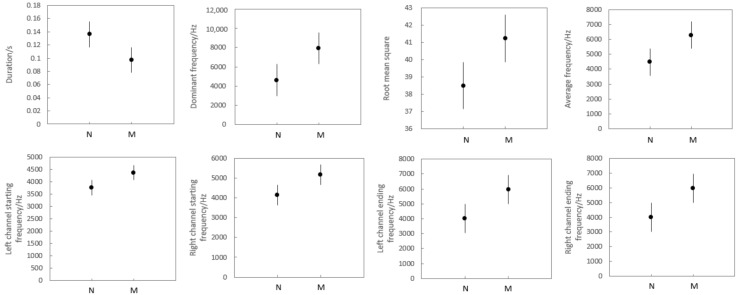
Average ± SE of two kinds of alarm call information. On all graphs: N = natural without disturbance; M = model disturbance. Sample sizes: N = 26; M = 10.

**Figure 7 animals-13-01271-f007:**
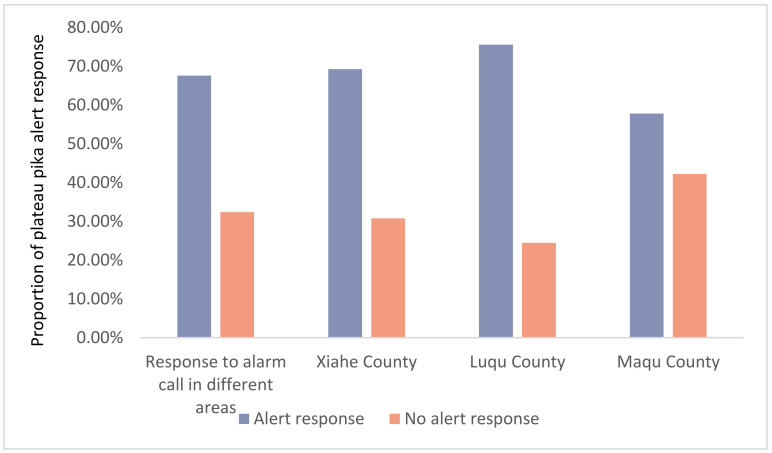
In these three areas, through analyzing the behavioral responses after playback of the alarm call (*p* > 0.05), it was found that there was no significant difference in the response of vigilance behavior.

**Figure 8 animals-13-01271-f008:**
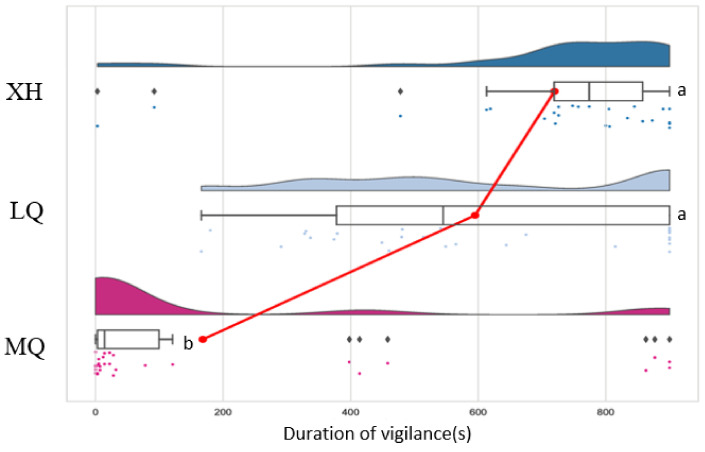
Alert duration of plateau pikas in different areas during playback experiments. Note: XH stands for Xiahe county (altitude: 3212 m), LQ stands for Luqu county (altitude: 3383 m) and MQ stands for Maqu county (altitude: 3587 m). The graph is composed of four parts, from top to bottom: a half violin plot representing the data core density; a boxplot representing the average level and fluctuation degree of the sample data; data points providing the dispersion of the sample data; and a line connecting the mean values of different groups. By comparing the warning duration at XH, LQ and MQ, it was found that their averages were 721.817 s, 594.929 s and 168.258 s, respectively. Note: Firstly, arrange all averages in descending order, and then label the letter a on the largest average; and compare the average with the following averages. If there is no significant difference, label it with the letter a until a significant difference is found, and label it with the letter b. Further statistical testing (*p* < 0.05) showed that there was a significant difference between the warning durations at the three locations and that the warning duration at XH and LQ was significantly higher than that at MQ.

**Table 1 animals-13-01271-t001:** Information regarding the study sites.

Study Site	Altitude (m)	Mean Annual Air Temperature (°C)	Mean Annual Precipitation (mm)	Dominant Plant Species
Bola township	3212	2.1	580.0	*Elymus nutans*, *Poa annua*, *Potentilla anserina*, *Kobresia graminifoli*, *Potentilla fragarioides*
Maai township	3383	2.9	592.5	*Kobresia pygmaea*, *Stipa aliena*, *Kobresia ansuensis*, *Festuca rubre*, *Polygonum viviparum*
Nima township	3527	1.8	593.3	*Elymus nutans*, *Stipa purpurea*, *Poa annua*, *Potentilla ansrina*, *Kobresia graminifolia*, *Anemone trullifolia*

## Data Availability

The datasets used and/or analyzed during the current study are available from the corresponding author upon reasonable request.

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
