# Peer review of "Alarm Calling in Plateau Pika (Ochotona curzoniae): Evidence from Field Observations and Simulated Predator and Playback Experiments"

_animals, 2023, doi:10.3390/ani13071271_

Round 1
Reviewer 1 Report
A very interesting experiment carried out in the field using modern methods. The presented research objective was fully achieved through the planned methodology.
Were picas inventories carried out in the study areas? If so, by what methods?
Were the animal populations similar in each of the three research areas? What was the density of animals per unit of area?
In my opinion, the weakest point of the methodology is determining whether tagged animals or other animals were recorded.
Another doubt is the choice of the predator.
Why didn't the authors decide to simulate a bird of prey using, for example, a drone?
The authors described the reaction of picas to the appearance of livestock: cattle and sheep on the pasture. What about a human? After all, there were also people tending farm animals!
Author Response
Thank you for your review of this article. Please see the attachment for detailed responses

Reviewer 2 Report
This study provides a more in depth examination of the vocalizations of the plateau pika and incorporates playback experiments to investigate the functions of their alarm calls. The authors use a good design for the study, but some more details are needed to clarify the methodology. The results also need a bit more clarification and I recommend the authors report more about of the acoustic parameters of the calls. Some of the calls are described with different terms (e.g., short/long calls, whistle, tweeting, chirp, etc.), but it is not clear if there are referring to one or more of the call types. Some of the content of the discussion seems outside the scope of this study, and I would recommend reorganizing it to focus more on the results and how they compare to other studies. I have included some more specific comments in the attached document.

Author Response
Thank you for your review of this article. Please see the attachment for detailed responses.

Round 2
Reviewer 2 Report
Thank you to the authors for submitting these revisions. The manuscript is much improved and I appreciate your explanations to my questions. I think there are still a few areas where some more details and clarity in the wording could help strengthen the findings. Please see my suggested edits in the attached document.

Author Response

(The authors gave the same response as above.)
